# Morphological and Molecular Characterization of *Pratylenchus dakotaensis* n. sp. (Nematoda: Pratylenchidae), a New Root-Lesion Nematode Species on Soybean in North Dakota, USA [note 1]

**DOI:** 10.3390/plants10010168

**Published:** 2021-01-17

**Authors:** Zafar A. Handoo, Guiping Yan, Mihail R. Kantor, Danqiong Huang, Intiaz A. Chowdhury, Addison Plaisance, Gary R. Bauchan, Joseph D. Mowery

**Affiliations:** 1Mycology and Nematology Genetic Diversity and Biology Laboratory, USDA, ARS, Northeast Area, Beltsville, MD 20705, USA; mihail.kantor@usda.gov; 2Department of Plant Pathology, North Dakota State University, Fargo, ND 58108, USA; guiping.yan@ndsu.edu (G.Y.); danqiong.huang85@yahoo.com (D.H.); intiaz.chowdhury@ndsu.edu (I.A.C.); addison.plaisance@ndsu.edu (A.P.); 3Electron & Confocal Microscopy Unit, USDA, ARS, Northeast Area, Beltsville, MD 20705, USA; gary.bauchan@usda.gov (G.R.B.); joseph.mowery@usda.gov (J.D.M.)

**Keywords:** D2/D3, description, *Glycine max*, lesion nematode, molecular, morphology, morphometrics, phylogeny, *Pratylenchus dakotaensis* n. sp., soybean, ITS, *COX1* gene, PCR-RFLP

## Abstract

Root-lesion nematodes (*Pratylenchus* spp.) of the genus *Pratylenchus* Filipjev, 1936, are among the most important nematode pests on soybean (*Glycine max* (L.) Merr.), along with soybean cyst and root-knot nematodes. In May 2015 and 2016, a total of six soil samples were collected from a soybean field in Walcott, Richland County, ND and submitted to the Mycology and Nematology Genetic Diversity and Biology Laboratory (MNGDBL), USDA, ARS, MD for analysis. Later, in 2019, additional nematodes recovered from a greenhouse culture on soybean originally from the same field were submitted for further analysis. Males, females, and juveniles of *Pratylenchus* sp. were recovered from soil and root samples and were examined morphologically and molecularly. DNA from single nematodes were extracted, and the nucleotides feature of three genomic regions targeting on the D2–D3 region of 28S rDNA and ITS rDNA and mitochondrial cytochrome oxidase subunit I (*COX1*) gene were characterized. Phylogeny trees were constructed to ascertain the relationships with other *Pratylenchus* spp., and polymerase chain reaction-restriction fragment length polymorphism (PCR-RFLP) was performed to provide a rapid and reliable differentiation from other common *Pratylenchus* spp. Molecular features indicated that it is a new, unnamed *Pratylenchus* sp. that is different from morphologically closely related *Pratylenchus* spp., including *P*. *convallariae*, *P. pratensis*, *P. fallax*, and *P*. *flakkensis*. In conclusion, both morphological and molecular observations indicate that the North Dakota isolate on soybean represents a new root-lesion nematode species which is named and described herein as *Pratylenchus dakotaensis* n. sp.

## 1. Introduction

The genus *Pratylenchus* Filipjev, 1936, is one of the most important nematode genera in terms of the economic impact they have on crops [1,2]. Currently, the genus contains approximately 100 species [3,4,5], with new species being described very frequently. The root-lesion nematodes are ranked as the third most important group of plant-parasitic nematodes after root-knot and cyst nematodes [2] in terms of economic loss in agriculture and horticulture. Frederick and Tarjan [6] published a compendium of the *Pratylenchus* genus in 1989 in which they reported 89 species. In 1989, Handoo and Golden [7] also published a key and compendium to 63 valid species, including an update of the work done by different workers on the genus. The plants reported as hosts for the genus are very large. For example, one species, *Pratylenchus penetrans*, has been reported to have more than 400 plants as hosts [8]. 

On soybean, root-lesion nematodes are one of the most damaging nematodes that feed on the soybean roots [9]. Two species, *Pratylenchus brachyurus* and *P. penetrans*, have been reported to cause damage to the roots of soybean plants [10]. For example, soybean plant growth was suppressed by *Pratylenchus brachyurus* nematodes, with a negative correlation being reported between the number of nodes on the main stem and the number of nematodes at planting [11]. Despite the fact that nearly 100 different species of *Pratylenchus* have been described to date, only 5 have been reported in North Dakota, namely the *P. agilis* [12], *P. neglectus*, *P. scribneri* [13,14], and 2 new species of *Pratylenchus* [15,16]. 

The objective of this study was to describe one of these two new species using light microscopy (LM) and scanning electron microscopy (SEM) observations and assess the diagnostic values of their morphological and molecular characters. The morphometric details of females and males were recorded and compared to closely related species. Also, the molecular details using ITS, 28S, and *COX1* sequences were obtained and compared to the existing information in GenBank. PCR-RFLP was performed to rapidly and reliably differentiate it from other important *Pratylenchus* spp. species. 

## 2. Materials and Methods

Nematode suspensions extracted from soil samples were sent to the MNGDBL, Beltsville, MD in May of 2015 and 2016. The origin of the soil samples was a field cultivated with soybean in Walcott, Richland County, ND. Nematodes were extracted from soil using the sugar centrifugal flotation method [17]. Each sample contained between 125 and 2000 root-lesion nematodes per kg soil [15]. In 2019, infested soil samples from the same field were planted to soybean cultivar Barnes in a greenhouse room at 22 °C. After 15 weeks of growth, the plants were harvested, and root-lesion nematodes were extracted from both the roots and soil using the Whitehead tray method [18]. Additional nematodes recovered from the greenhouse culture on soybean were submitted to the MNGDBL, Beltsville, MD for further analysis.

### 2.1. Morphological Examination

Females and males were recovered from the root and soil samples using the Whitehead tray method extraction method [18]. Nematodes were fixed in 3% formaldehyde and processed with glycerin by the formalin glycerin method [19,20]. Photomicrographs of females and males were made with an automatic 35 mm camera attached to a compound microscope with an interference contrast system, and light microscopic images of fixed nematodes were taken on a Nikon Eclipse Ni compound microscope using a Nikon DS-Ri2 camera. Measurements were made with an ocular micrometer on a Leica WILD MPS48 Leitz DMRB compound microscope. All measurements are in micrometers unless otherwise stated. 

For the Low-Temperature Scanning Electron Microscopy (LT-SEM), nematodes were observed using the techniques described by Carta et al. [21], Kantor et al. [22], and Handoo et al. [23]. 

### 2.2. DNA Extraction, PCR, and Sequencing 

DNA was extracted from a single individual nematode using the Proteinase K method [24]. Briefly, the chopped nematode pieces were transferred into a 0.5 mL sterile Eppendorf tube containing 2 µL of 10 × PCR buffer with MgCl_2_, 2 µL of 600 µg/mL Proteinase K (Roche, Indianapolis, Indiana), and 6 µL of distilled ddH_2_O. Tubes were incubated at −20 °C for at least 30 min followed by 65 °C for 1 h and 95 °C for 10 min. DNA samples from 3 nematode individuals were prepared, which represented 3 biological replicates. 

Nucleotide sequences of D2D3 fragment of 28S rDNA and ITS rDNA regions from ribosomal DNA and *COX1* (cytochrome oxidase subunit I) gene from mitochondrial DNA were obtained by either direct sequencing using purified PCR products or T-A cloning. For D2–D3 region of 28S rDNA, the primer set of D2A (5′-ACAAGTACCGTGAGGGAAAGTTG-3′) and D3B (5′-TCGGAAGGA ACCAGCTAC TA-3′) was used [25]. For ITS rDNA, the primer set of 18S (5′-TTGATTACGTCCCTGCCCTTT-3′) and 26S (5′-TTTCACTCG CCGTTACTAAGG-3′) was used [26]. For *COX1* gene, the primer set of JB3 (5′-TTTTTTGGGCATCCTGAGGTTTAT-3′) and JB4.5 (5′-TAAAGAAAGAACATAATGAAAATG-3′) was used [27]. The PCR were set up on Bio-Rad T100 Thermal Cycler (Hercules, CA, USA) as recommended [25,26,27]. For direct sequencing (D2–D3 region of 28S rDNA and *COX1* gene), PCR products were purified using Bio-tek E.Z.N.A. Cycle-Pure Kit (Omega, Norcross, GA, USA) and then sent to Genscript for sequencing (Genscript, Piscataway, NJ, USA). For cloning and sequencing, target PCR products were segregated on a 1.0% agarose gel, purified using Gel Extraction Kit (Omega), and cloned into pGEM-T Vector using pGEM-T Vector System II Kit (Promega, Madison, WI, USA) according to the manufacturer’s instructions. Plasmid DNA was then extracted from the white colonies grown on indicator plates containing X-gal and IPTG, using a PerfectPrep™ Spin Mini Kit (5 PRIME Inc., Gaithersburg, MD, USA), and sent to Genscript for sequencing. Three sequences were generated for each of the three target areasand the corresponding consensus sequences of D2–D3 region of 28S rDNA, ITS rDNA, and *COX1* gene were deposited into GenBank to obtain their accession numbers (MW290216.1 for D2–D3, MW290217.1 for ITS, and MW309316.1 for *COX1*). 

### 2.3. Phylogenetic Analysis 

Phylogenetic relationships among *Pratylenchus* spp. were analyzed using Maximum Likelihood (ML) method of MEGA7 software [28]. Available DNA sequences of the 28S rDNA, ITS rDNA, and *COX1* gene of other *Pratylenchus* spp. species were retrieved from the NCBI Nucleotide Database (https://www.ncbi.nlm.nih.gov/nucleotide) and aligned with corresponding sequences obtained in this study by MUSCLE software (v3.8.31) with default settings for the highest accuracy. The gaps and missing data were completely removed. Accordingly, the General Time Reversible model was selected using gamma distribution, with invariant sites and 5 gamma-distributed rate categories to account for rates and patterns. Finally, the phylogeny trees were constructed using maximum likelihood method with 1000 bootstrap replications. 

### 2.4. PCR-RFLP Analysis

To differentiate different *Pratylenchus* spp., polymerase chain reaction-restriction fragment length polymorphism (PCR-RFLP) analysis was performed using ITS rDNA amplified by the primers 18S/26S [26,29] with restriction endonucleases *Hind III* and *Hha I*. The PCR was carried out in a 20 µl reaction comprising of 1.5 µL DNA template, 0.4 µM forward and reverse primers, 0.2 mM dNTP, 1.5 mM MgCl_2_, 1 × Green GoTaq^®^ Flexi buffer, and 1 U GoTaq^®^ Flexi DNA Polymerase (Promega, Madison, WI, USA) with conditions of pre-denaturing at 94 °C for 3 min followed by 35 cycles of 94 °C for 1 min, 55 °C for 1 min, and 72 °C for 1 min, with a final extension at 72 °C for 10 min. The digestion was performed in 20 µL reaction mixtures containing 5U restriction enzyme, 1× RE buffer, 2 µg acetylated BSA, and 10 µl PCR products at 37 °C for 2 h. The digested fragments were separated in 2 % agarose gel at 100 volts (V) for 20 min. The gel was visualized under UV light and images were captured using an AlphaImager^®^ Gel Documentation System (Proteinsimple Inc., Santa Clara, CA, USA).

## 3. Results

### 3.1. Systematics 

Pratylenchus dakotaensis n.sp.


http://zoobank.org/urn:lsid:zoobank.org:pub:F89CA839-1A5B-4A27-BA53-8F8D6633E89C


(Figure 1, Figure 2, Figure 3 and Figure 4, Table 1). 

#### 3.1.1. Measurements

#### 3.1.2. Description

*Female*: Slender and vermiform body, assuming straight or arcuate form when killed by gentle heat and tapering at both ends. Lateral field with four lines, with the outer two lines being areolated more so at tail region. Occasionally, additional oblique lines are noted in between the two inner lines. The lip region is flat to rounded or dome-shaped, slightly offset with the body contour and bearing three fine annuli. The en face view shows a divided face with rectangular subdorsal and subventral lips fused with oral disc in a dumbbell- to dome-shaped pattern that is separated from lateral lip sectors by three almost straight, often obscure incisures forming an obtuse angle. The stylet is short and robust with rounded knobs. The distance of the dorsal pharyngeal gland orifice to the stylet base is 2–3 µm. The procorpus is generally cylindrical but narrows near the middle and at junction with median bulb. The median bulb is muscular and rounded to slightly oval-shaped with cuticularized valve plates. The nerve ring encircles median part of isthmus. The excretory pore is located posterior to the nerve ring. The hemizonid is located at the two annuli anterior to excretory pore. The pharyngeal glands’ nuclei are in tandem, elongate, and overlapping with the intestine ventrally. The reproductive system is monodelphic, prodelphic, with the ovary outstretched with single row of oocytes. The post-uterine sac is 18–20 µm long, and the vulva is located 78–83% of total body length from anterior end. The vulval lips are slightly protruding with no lateral flaps and epiptygma. The tail broad is conical, with 16–26 narrow irregularly annuli with terminal annuli usually wider than other tail annuli. The tail terminus is distinctly crenate/annulated with rounded to truncate- or clavate-shaped. The phasmids are prominently located at approximately the middle of the tail. 

*Male*: Males are common and are similar to females, including the lip region, except for the sexual dimorphism. The stylet slightly is shorter than females, measuring 15.5 µm to 16.5 µm long. The lateral fields have four incisures, with the outer two lines mostly areolated. The reproductive system is composed of a single testis, which is anteriorly outstretched. The spicules and gubernaculum are ventrally curved, measuring 16–18.5 µm and 4–5 µm, respectively. The tail is short, bluntly rounded to pointed. The bursa encircle the entire tail. The ventral surface of the bursa is coarsely annulated. The phasmids are prominent.

### 3.2. Type Host and Locality

*Pratylenchus dakotaensis* n. sp is associated with roots and around soil from a soybean field in Richland County, ND. The global positioning coordinates for Richland County are 43.188221° N and 124.390174° W. 

### 3.3. Type Material

Holotype (female): Slide T-740t, deposited in the United States Department of Agriculture Nematode Collection, Beltsville, MD, USA. Paratypes (Females, and Males): Same data and repository as holotype, Slides T-7153p to T-7158p. Additional females on slide numbers T-7159p at University of California, Riverside, CA, USA, and T-7160p at Fera, Plant Pest Disease Cultures and Collections, York, United Kingdom. 

### 3.4. Diagnosis and Relationships

*Pratylenchus dakotaensis* n. sp. is characterized by a combination of the following morphological features in females: Slender, vermiform body, assuming straight or arcuate form, lateral field with four lines, with the outer two lines being areolated; the lip region is flat to rounded or dome-shaped, slightly offset with the body contour and bearing three fine annuli; the en face view shows a divided face with a rectangular subdorsal and subventral lips fused with the oral disc in a dumbbell- to dome-shaped pattern; the stylet is short and robust, with rounded knobs; the vulva is located at 78–83% of total body length from anterior end; the vulval lips are slightly protruding with no lateral flaps and epiptygma; the tail is broad and conical, with 16–26 narrow irregularly annuli, and the terminal annuli usually wider than the other tail annuli; the tail terminus is distinctly crenate/annulated with rounded to truncate- or clavate-shaped. Males are common; their stylet is slightly shorter than females; the spicules and gubernaculum are ventrally curved, measuring 16–18.5 µm and 4–5 µm, respectively; the tail is short, bluntly rounded to pointed; and the bursa encircle the entire tail. 

*Pratylenchus dakotaensis* n. sp. is morphologically closely related to *Pratylenchus convallariae*, *P. pratensis*, and *P. fallax*. Sequence (GenBank accession No. MW290216, 702 bp) from the 28S D2–D3 had less than 94.2% similarity with these three species. In addition, it had 100% identity with *Pratylenchus* sp. (MN251269) from Lafayette County, Wisconsin and 98.6% identity with *P. scribneri* (MG925218) from Ohio, USA. The ITS sequence (GenBank accession No. MW290217, 1226 bp) of *P. dakotaensis* had less than 93.1% similarity with other *Pratylenchus* spp. including *P*. *convallariae*, *P. pratensis*, *P. fallax*, *P. scribneri*, and many isolates of an unknown *Pratylenchus* sp. Sequence (GenBank accession No. MW309316, 419 bp) from *COX1* gene had 97.5% identity with five isolates of a *Pratylenchus* sp. from Atchison County, Kansas, USA and less than 84.6% identity with other *Pratylenchus* spp. Thus, the sequence data did not support *P*. *convallariae*, *P. pratensis*, or *P. fallax*. Another morphologically closely related species is *P. flakkensis*, but *P. dakotaensis* differs from *P*. *flakkensis* in several morphological characters, with a high head, three head annuli, slight longer stylet in females, higher vulva percentage, and longer spicule in males. Accordingly, both morphological and molecular observations with the known and abovementioned closely related species indicate that the North Dakota isolate on soybean represents a new root-lesion nematode species, which is described herein as *Pratylenchus dakotaensis* n. sp. 

### 3.5. Etymology

The species name was derived from North Dakota, the geographic origin. 

### 3.6. Molecular Analysis

Phylogenetic relationships based on the D2–D3 region of 28S rDNA, ITS rDNA, and *COX1* gene were generated using the Maximum Likelihood method using corresponding nucleotides from *Pratylenchus* species (Figure 4, Figure 5 and Figure 6). In the tree constructed using the D2–D3 region of 28S rDNA, which is considered as the most evolutionally conserved region, *P. dakotaensis* was more likely closely related with *P*. *scribneri*, *P. hexincisus*, *P*. *pseudocoffeae P. loosi*, *P. speijeri, P*. *coffeae,* and *P*. *hippeastri* (ML = 90), compared with those morphological closely related species, including *P*. *convallariae*, *P. pratensis*, *P. fallax*, and *P*. *flakkensis*. Similarly, in the trees constructed using ITS rDNA and *COX1* gene, *P. dakotaensis* was also clustered with those closely related species in the tree of 28S rDNA. 

The RFLP analysis using the ITS rDNA region was performed to distinguish *P. dakotaensis* n. sp. from other common, important root-lesion nematode species (Figure 7). The results revealed that PCR products from the ITS region with two digestion enzymes (*Hind III* and *Hha I*) generated the same banding pattern for nine samples from the field infested with *P. dakotaensis* n. sp. but different banding patterns from *P. scribneri, P. neglectus,* and *P. penetrans*, which are the major *Pratylenchus* species in the region. 

## 4. Discussion

Based on the molecular results obtained using the 28S D2–D3 primers, the North Dakota population had less than 94.2% similarity with morphologically closely related *Pratylenchus* spp., including *P*. *convallariae*, *P. pratensis*, *P. fallax*, and *P*. *flakkensis*. After analyzing the molecular data obtained by sequencing the ITS region, less than 93.1% similarity with *P*. *convallariae*, *P. pratensis*, and *P. fallax* was observed. The sequence from the *COX1* gene had less than 84.6% identity with other *Pratylenchus* spp., except 97.5% identity with five isolates of an undefined *Pratylenchus* sp. Looking at the morphometric data, the population of *P. dakotaensis* is similar to *P*. *flakkensis*. Despite the similarities between the two, several differences have been observed, such a high head in the North Dakota population, three head annuli instead of two, slightly longer stylet in females, higher vulva percentage, and longer spicule in males. In conclusion, combining all morphological and molecular data and observations with the known and abovementioned closely related species indicates that the North Dakota isolate on soybean represents a new root-lesion nematode species, described here as *Pratylenchus dakotaensis* n. sp. Interestingly, the 28S D2–D3 sequence of an unknown *Pratylenchus* sp. from Wisconsin, USA (GenBank accession No. MN251269) showed 100% identity with this new species. The specimens from Wisconsin and North Dakota need to be compared thoroughly to determine whether the Wisconsin population belongs to *Pratylenchus dakotaensis* n.sp.

## Figures and Tables

**Figure 1 plants-10-00168-f001:**
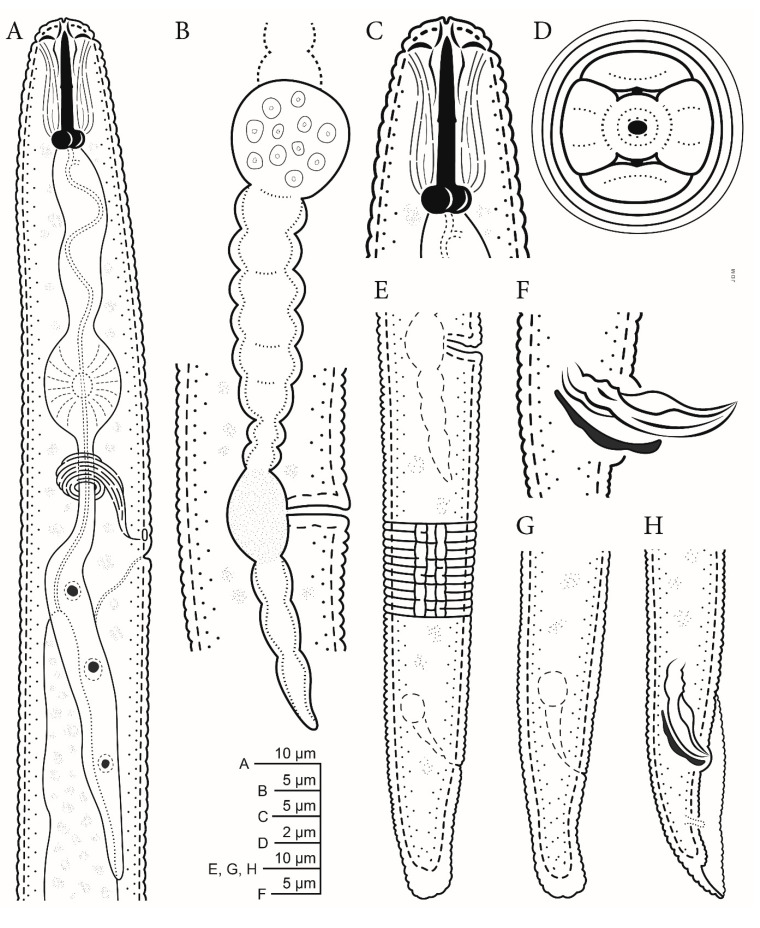
Line drawings of *Pratylenchus*
*dakoaiensis* n. sp.: (**A**) Female pharyngeal region; (**B**) vulval region showing vulva, uterus, and spermatheca, (**C**) female lip region showing stylet; (**D**) details of the lip region showing the oral disc (*en face view*); (**E**,**G**) female tails with E showing lateral field with four lines; (**F**,**H**) male tails showing spicules and gubernaculum.

**Figure 2 plants-10-00168-f002:**
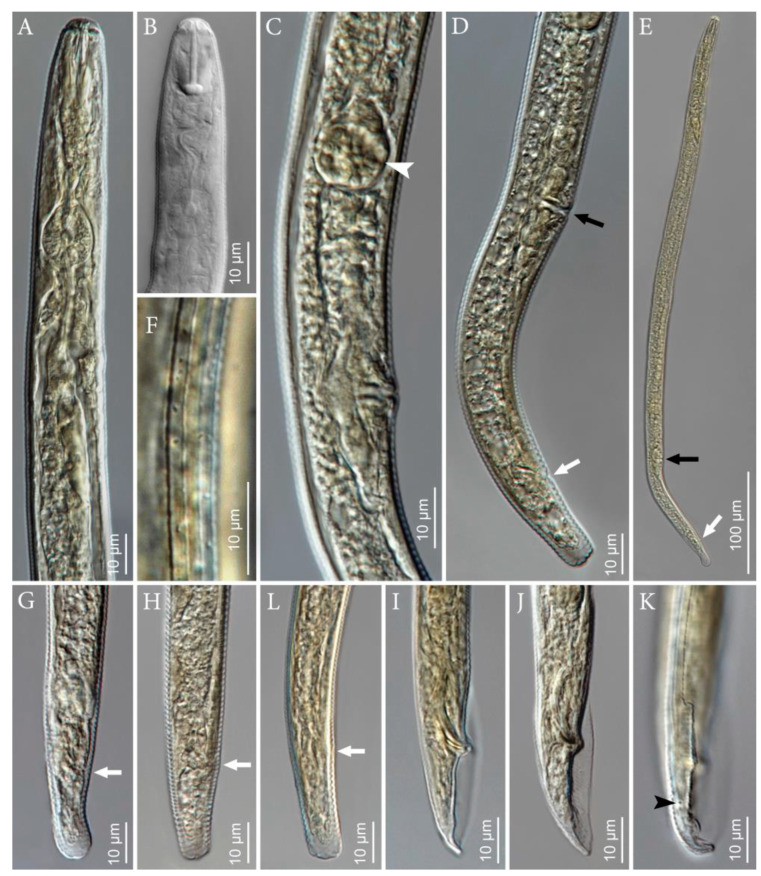
Photomicrographs of *Pratylenchus dakotaensis* n. sp.: (**A**) Female anterior end showing pharyngeal region; (**B**) female anterior end showing stylet; (**C**) female vulval area with arrow pointing the spermatheca; (**D**) female posterior end with arrows in black and white pointing to vulval and anal openings, respectively; (**E**) entire female with arrows in black and white pointing to vulval and anal openings; (**F**) female mid body showing lateral field with four lines; (**G**,**H**,**L**) female posterior ends showing tail variations and arrows pointing to anal areas; (**I**,**J**) male posterior ends showing spicule and bursa; (**K**) male posterior end with arrow pointing the phasmid.

**Figure 3 plants-10-00168-f003:**
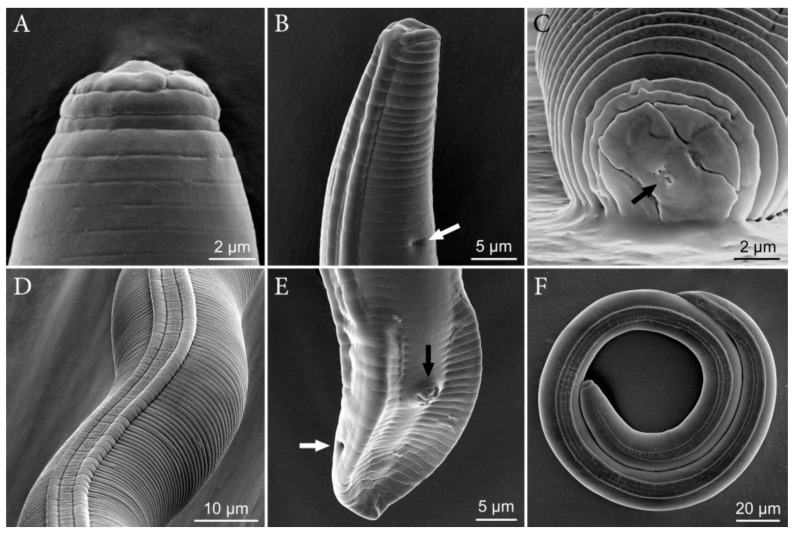
SEM images of *Pratylenchus dakotaensis* n. sp.: (**A**) Male specimen, head; (**B**,**C**) female posterior and anterior ends, (**B**) female posterior end, arrow showing anal opening, and (**C**) arrows showing oral opening; (**D**) female mid-body region showing lateral field; (**E**) male posterior end arrows in white and black showing cloaca opening and spicule, respectively; (**F**) whole specimen lateral field.

**Figure 4 plants-10-00168-f004:**
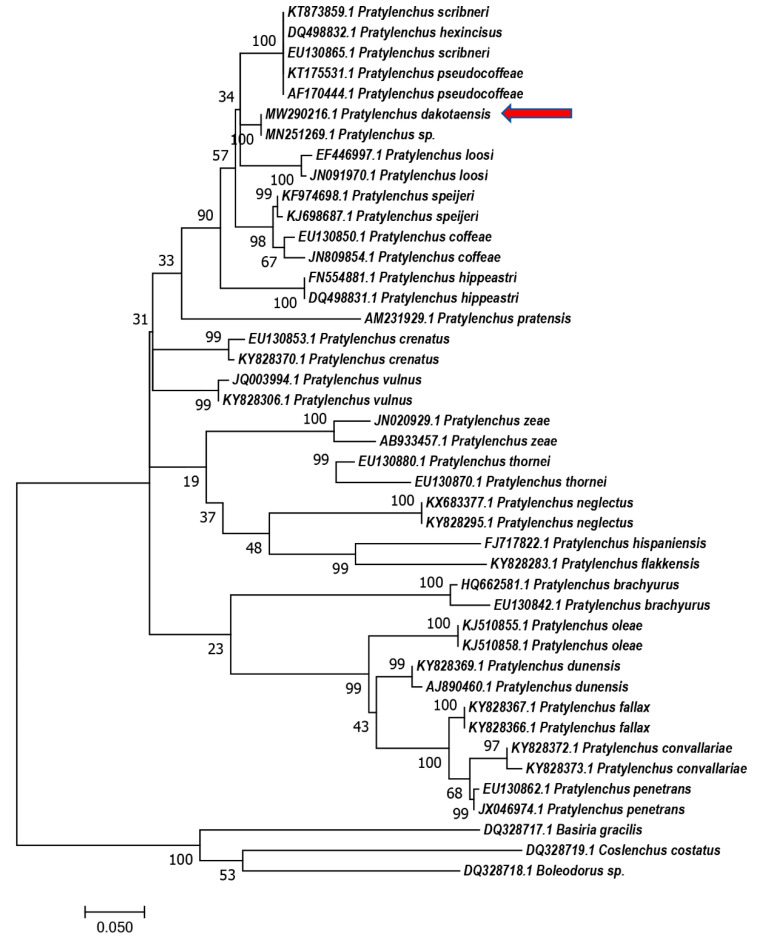
Phylogenetic relationships of *Pratylenchus dakotaensis* n. sp. (red arrow) from D2–D3 28S with related *Pratylenchus* spp. sequences from GenBank based on Maximum Likelihood analysis. Support values are given above branches. *Coslenchus costatus*, *Boleodorus* sp., and *Basiria gracilis* were served as outgroups.

**Figure 5 plants-10-00168-f005:**
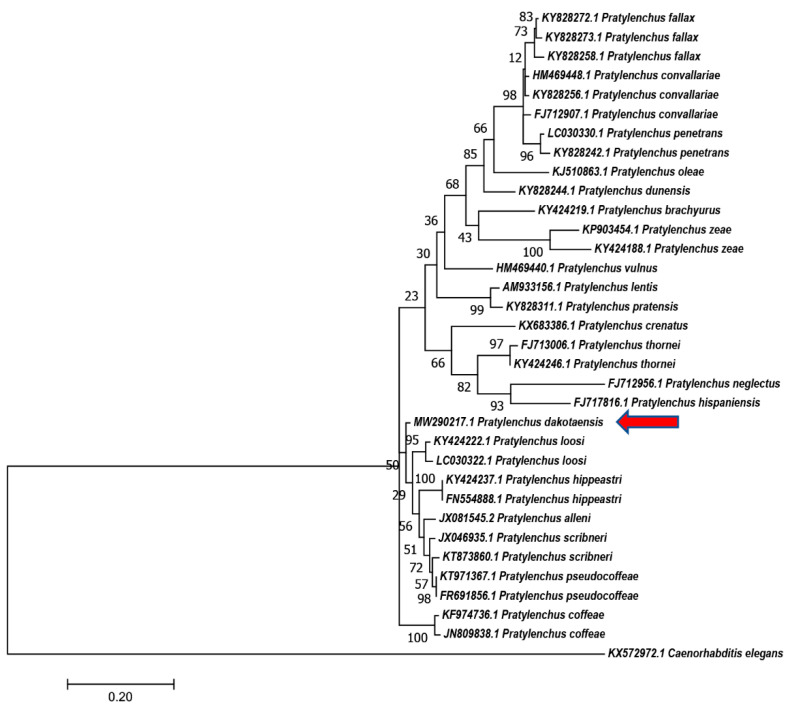
Phylogenetic relationships of *Pratylenchus dakotaensis* n. sp. (red arrow) from ITS rDNA with related *Pratylenchus* spp. sequences from GenBank based on Maximum Likelihood analysis. Support values are given above branches. *Caenorhabditiss elegans* was served as an outgroup.

**Figure 6 plants-10-00168-f006:**
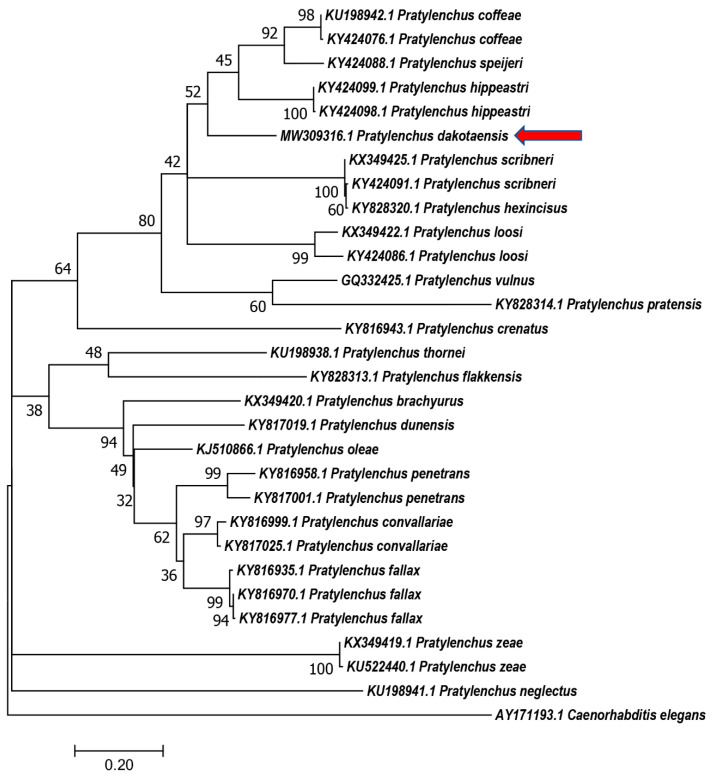
Phylogenetic relationships of *Pratylenchus dakotaensis* n. sp. (red arrow) from partial cytochrome oxidase subunit I (*COX1*) gene with related *Pratylenchus* spp. sequences from GenBank based on Maximum Likelihood analysis. Support values are given above branches. *Caenorhabditiss elegans* was served as an outgroup.

**Figure 7 plants-10-00168-f007:**
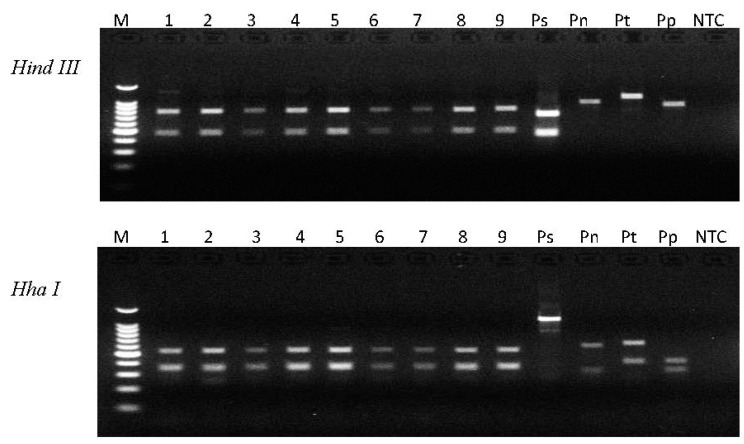
PCR-RFLP of *Pratylenchus* spp. using *Hind III* and *Hha I* enzymes. PCR products were amplified with primers rDNA2-V/PnSeqR targeted on ITS rDNA. The letter M refers to 100 bp DNA ladder from Promega, lanes 1–9 were nematode DNA extracted from single individuals isolated from the field having infestation of *Pratylenchus dakotaensis* n. sp., Ps refers to DNA from *P. scribneri* (ND), Pn refers to DNA from *P. neglectus* (ND), Pt refers to DNA from *P. thornei* (OR), Pp refers to DNA from *P. penetrans* (MN), and NTC refers to no-template control using ddH_2_O instead of nematode DNA in the PCR reaction.

**Table 1 plants-10-00168-t001:** Morphometrics of *Pratylenchus dakotaensis* n. sp. All measurements are in µm and in the form: mean ± standard deviation (s.d.) (range).

Character	Holotype	Females	Males
n		22	7
L	552.0	484.5 ± 39.9 (390.0–555.0)	445.7 ± 56.0 (355.0–502.0)
a	27.6	23.4 ± 2.8 (20.8–29.8)	23.7 ± 2.01 (20.8–25.2)
b	4.2	4.0 ± 0.4 (3.2–4.8)	4.06 ± 0.5 (3.2–4.8)
c	22.0	20.2 ± 1.7 (16.8–24.1)	20.0 ± 1.7 (16.7–21.3)
C’	2.0	1.9 ± 0.3 (1.4–2.4)	2.1 ± 0.17 (1.9–2.4)
Anal body width	12.0	13.0 ± 2.0 (10.0–16.0)	10.9 ± 0.7 (10.0–12.0)
V%	80.0	80.2 ± 1.5 (78.0–83.0)	-
Maximum body width	21.0	21.9 ± 2.5 (20.8–29.8)	18.9 ± 1.5 (17.0–21.0)
Stylet length	16.0	17.5 ± 0.3 (16.0–18.0)	16.0 ± 0.3 (15.5–16.5)
Distance from head end to posterior end of esophageal glands	130.0	118.8 ± 9.7 (110.0–140.0)	109.3 ± 5.7 (101.0–115.0)
Tail length	25.0	24.4 ± 2.4 (20.0–30.0)	23.1 ± 1.8 (20.0–25.0)
Spicule length	-	-	17.5 ± 0.82 (16.0–18.5)
Gubernaculum	-	-	4.5 ± 0.5 (4.0–5.0)


## Data Availability

The data presented in this study are openly available in Plants at https://doi.org/10.3390/plants10010168.

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
