# Peer review of "Morphological and Molecular Characterization of Pratylenchus dakotaensis n. sp. (Nematoda: Pratylenchidae), a New Root-Lesion Nematode Species on Soybean in North Dakota, USAâ€"

_plants, 2021, doi:10.3390/plants10010168_

Round 1
Reviewer 1 Report
The manuscript included all the suggestions raised by me and deserves final acceptance in Plants.
Reviewer 2 Report
I have no more/new comments or questions regarding the current revised version of manuscript
This manuscript is a resubmission of an earlier submission. The following is a list of the peer review reports and author responses from that submission.
Round 1
Reviewer 1 Report
I had a oportunity review the manuscript entitled "Morpological and molecular characterization of Pratylenchus dakotiensis n.sp. ....." submited to Plants. The manuscript has standard structuring of new species described studies, contains all required methodological principles which suggest that P. dakotiensis can be accepted as new species of the genus Pratylenchus. Several comments are included in revised file.

Reviewer 2 Report
Nematologically, the manuscript follows a standard format for a taxonomic description. The morphological aspect of the manuscript is of a high quality (especially the line drawings and photomicrogrpahs). In contrast, the molecular aspect is poor and there appears to be a significant lack of understanding of how to interpret phylogenetic trees. Also, methodological information is omitted (for example, how many sequences were generated for each of the three target areas? Surely it was not the three as stated on lines 105-107?).
All three phylogenetic trees (Figures 5-7) are poorly resolved, i.e. bootstrap values for several key nodes are exceptionally low thus the listed sequences mostly do not form the relationships as depicted. Normal convention is that bootstraps > 70% are considered robust and more robust the closer to 100%. Bootstraps < 70% are viewed as not being phylogenetically relevant. Common practice is to provide a ML (included in the manuscript) and a Bayesian analysis - why was a Bayesian analysis not included in this manuscript? The poorly resolved trees does not support the statement on lines 247-248.
Further, there is confusion as to the selection criteria for the sequences included in the phylogenetic trees. For example, those compared in the text (lines 208-214) are not included in the trees - why are they omitted? It is stated on lines 208-212 that the newly described species is morphologically closely related to P. convellariae, P. pratensis and P. fallax yet P. pratensis is not included in the phylogenetic trees. Why are the stated accession numbers of P. dakotiensis used as a comparator (lines 208-214) different to those displayed in Figures 5-7?
As the authors have sequence data from three target DNA regions it makes no sense to include a RFLP analysis. What was the scientific rationale to include RFLP data when the authors could simply compare sequence data which is infinitely more accurate than RFLP.
There are numerous errors and a lack of clarity throughout the manuscript. For example, it is stated (line 191) that the holotype is female. If that is true why do the values for c, anal body width and maximum body width fall outside the range of the morphometrics for the 22 females assessed during the study? Staying with Table 1, there are numerous inconsistencies with the presented data with errors in range (e.g. female c) and an inconsistent range of resolution of the data (0 to 2 decimal points). There are too many errors to list but other main errors to note include -
poor justification of the importance of Pratylenchus - was the genus included in Jones et al which listed the 10 most important global plant parasitic genera?
a lack of a stated comparator for the statement made on lines 44-45
assumed missing text as there is an apparent disconnect between lines 53 & 54
incorrect figure cited on line 63
missing reference for Baermann funnel
no explanation as to why there was a 3-4 year gap from soil sample receipt to glasshouse experiment
how does the DNA extraction method align with the newly published paper by Orlando et al in Nematology?
why were the Vrain primers known to be inefficient for Tylenchids used?
how many individuals were sequenced?
manuscript appears incomplete - see author contribution, funding and conflicts of interest sections
references need considerable work including an inconsistent use of full and abbreviated journal titles, the latter of which mostly are incorrect abbreviations
Reviewer 3 Report
This is a very interesting and novelty manuscript dealing with a new species of Pratylenchus. In my opinion the paper deserves publication in Plants with only minor revision, that may improves the manuscript. Please see below my suggestions:
- Italize Pratylenchus, Glycine max and other Pratylenchus species referred in abstract section
- Please, specify in materials and methods section if specimens were collected from inide roots of soybean, since you are dealing with a migratory endoparasite, and it is important confirm the thos plant is host of the nematode.
-
I suggest to authors register the new species in zookbank and include the accesion link in the manuscript.
Please also check, if the epithet name for Dakota is better dakotaensis......
- Please replace head by lip region throughout the manuscript.
-
Please, attention to some misspelling i.e. L208 convellariae instead of convallariae....
In relationships section, I suggest to authors compare the new species with another Pratylenchus species, see Nematology, 2010, Vol. 12(3), 429-451
L214, please provide accession number for COI.
I can see that authors only provide a single sequence for each molecular marker, should be better in the future to increase this number to at least 3 specimens in order to define the intraspecific diversity for each marker....
- Please replace annules by annuli
- L277-277, please provide the accession number of the coincident Pratylenchus from Wisconsin, and include in the phylogenetic tree.......
- I suggest to authors compare their results on phylogeny of the three markers with other previous paper, i.e. Eur J Plant Pathol (2016) 145:973–998, European Journal of Plant Pathology 140:053-067, Molecular Phylogenetics and Evolution 48 (2008) 491–505,
- Authors contribution is missing....